# Feasibility and acceptability of a tailored health coaching intervention to improve type 2 diabetes self-management in Saudi Arabia: a mixed-methods randomised feasibility trial

Abdullah N Almulhim [1,2] Atheer Alhowaish,[3] Alaa Madani,[3] Munirah AlQaddan,[3] Abdulaziz S Altuwalah,[3] Elizabeth Goyder[1]

¹School of Medicine and Population Health, The University of Sheffield, Sheffield, UK
²Public Health Department, College of Health Sciences, Saudi Electronic University, Riyadh, 13316, Saudi Arabia
³Health Education Administration, King Fahad Medical City, Riyadh, 12231, Saudi Arabia

**Correspondence to**
Mr Abdullah N Almulhim;
aalmulhim@seu.edu.sa

## ABSTRACT

**Background** Around one-third of the population of Saudi Arabia have been diagnosed with type 2 diabetes, a condition often requiring lifestyle changes. Personalised health coaching, a strategy developed to assist individuals in overcoming challenges to adopt healthy behaviours, has not yet been widely applied in the country.

**Aims** We aim to explore the feasibility and acceptability of tailored health coaching in Saudi Arabia, in order to help those with type 2 diabetes to more effectively manage their condition.

**Methods** Using a mixed-methods approach, this research involved a randomised controlled trial with 30 Saudi adults who have type 2 diabetes. They were randomly allocated into either the intervention or control arm for 12 weeks. The Capability, Opportunity, Motivation and Behaviour framework was used to guide the intervention implementation along with the Behaviour Change Techniques Taxonomy V.1. The primary goal was to assess the suitability and duration of the intervention, recruitment, retention and completion rates. The secondary outcome focused on the preliminary efficacy of the health coaching measured by the glycaemic index, blood pressure, body mass index (BMI), waist circumference, weight, patient self-efficacy and diabetes self-management.

**Results** The results showed high rates of eligibility, recruitment and retention (a screening rate of 90%, a recruiting rate of 79% and a retention rate of 97%). Notable improvements were observed in the health coaching group across five outcomes: haemoglobin A1c, BMI, waist circumference, patient self-efficacy and diabetes self-care. Qualitative findings highlighted the participants' perceived benefits from the intervention, including enhanced motivation, better understanding of diabetes management and a supportive coaching relationship. Participants expressed high satisfaction with the intervention and advocated for its expansion.

**Conclusion** The findings demonstrated positive outcomes, supporting the need for a larger randomised controlled trial to evaluate the efficacy of health coaching in improving diabetes self-management among individuals with type 2 diabetes in Saudi Arabia.

## STRENGTHS AND LIMITATIONS OF THIS STUDY

⇒ First health coaching intervention in Saudi Arabia.
⇒ First use of the Capability, Opportunity, Motivation-Behaviour model and Behavioural Change Techniques Taxonomy in a health coaching intervention.
⇒ Adherence to Consolidated Standards of Reporting Trials 2010 progression criteria.
⇒ Comprehensive impact assessment through mixed methods.
⇒ Limited generalisability due to specific settings and demographics.

## INTRODUCTION

Type 2 diabetes mellitus (T2DM) is a chronic illness that is becoming more common globally and represents a significant public health concern. Lifestyle factors play a crucial role in the development of this disease, with obesity often manifesting as a symptom of other lifestyle factors such as physical inactivity.[1] T2DM can lead to complications such as cardiovascular disease, kidney failure, nerve damage, and ocular and auditory problems, significantly increasing the risk of heart disease and stroke in people with diabetes.[2]

Managing diabetes can be a challenging task that encompasses various aspects of life, including social, economic and healthcare domains.[3] To help patients with T2DM effectively manage their condition, prioritising self-management is crucial.[4] T2DM is a prevalent chronic disease in Saudi Arabia (SA), with one of the highest rates in the Middle East and the seventh-highest globally.[5] Unhealthy diets, inactive lifestyles and rapid urbanisation are all contributing factors to this concerning trend.[6]

Despite the national efforts to control T2DM in the SA context, the disease continues to pose significant challenges.[7] While awareness campaigns are being employed, there is a notable difficulty among most T2DM patients in adopting healthier diets and lifestyles.[7] To address this issue, it is necessary to implement patient-centred approaches that prioritise the development of self-management skills rather than relying solely on education-based programmes.[8]

Health coaching, derived from various disciplines and behaviour change theories, has emerged as a client-centred approach gaining recognition in the field of health promotion. Although it has been widely explored in the literature and has shown promising results, this personalised one-on-one intervention has not yet been implemented within the healthcare systems of SA. Studies have demonstrated that health coaching, centred around patients' values, may offer additional benefits compared with traditional T2DM education programmes.[9] By focusing on enhancing patients' self-efficacy and essential skills such as goal-setting and problem-solving, health coaching can prove highly effective and financially viable.[10] Its successful application in diverse contexts has shown it to be a valuable tool in improving diabetes self-management.[11 12]

However, the effectiveness of existing health coaching interventions exhibits inconsistency due to variations in methodology, duration and intervention content.[11] Recent systematic reviews of randomised controlled trials (RCTs) employing health coaching have reported mixed results, with some studies suggesting its effectiveness while others claim it to be ineffective.[11 12] There is no agreed on health coaching intervention model, which creates uncertainty regarding the most effective strategies to adopt, particularly in the SA context. The inconsistent findings across studies may be due to the lack of agreement on the active ingredients and content of successful health coaching interventions.[11] Additionally, cultural differences between SA and Western countries highlight the need for tailored interventions to ensure that proposed health coaching models are appropriate and relevant.[13]

This feasibility study aims to assess the usability and acceptability of a health coaching intervention for T2DM self-management in SA, with the ultimate goal of reducing haemoglobin A1c (HbA1c) levels and informing a future large-scale RCT.

## MATERIALS AND METHODS
This research was reported in accordance with the Consolidated Standards of Reporting Trials (CONSORT) extension guidelines specific to pilot and feasibility RCTs.[14] A comprehensive protocol for this study has already been published.[15]

### Design
This is a randomised two-arm feasibility trial evaluating a 3-month intervention. The study adopted a mixed-methods approach, enabling the collection of both quantitative and qualitative data. An RCT design to address both evaluate the feasibility of RCT methods for recruitment, delivery and data collection, as well as a valid comparison of health coaching's effectiveness between the groups.[16] Integral to our approach was the implementation of the Behaviour Change Wheel (BCW) framework, used in adapting the health coaching intervention for the SA context. The BCW informed our intervention design in several key ways: First, by identifying suitable intervention functions from the Capability, Opportunity, Motivation-Behaviour (COM-B) model, we tailored the health coaching to the specific needs and barriers of our target population. Second, we used the Behaviour Change Technique Taxonomy V.1 to code the active ingredients of the intervention, mapping these to the BCW framework for alignment with established behaviour change principles.[17] Furthermore, a behavioural analysis, guided by the Theoretical Domains Framework functions and behavioural change techniques (BCTs), was conducted to address pre-identified barriers (see online supplemental table 1). This analysis led to the selection of 36 relevant BCTs, which were mapped to directly target the behavioural barriers in adults with poorly managed T2DM, our intervention's target population (see online supplemental table 2). By integrating the BCW, we were able to define the problem behaviours and target behaviours for intervention, ensuring a comprehensive and theory-driven approach as advised by BCW guidance.[18] Further information on the intervention can be found in a prior publication.[15]

### Patient and public involvement
Patients and the public were not involved during the process of this research.

### Intervention development
There have been various identified target behaviours that would result in the adoption of a healthy lifestyle and diet as recommended by UK lifestyle guidelines.[19] As of now, there are no evidence-based guidelines in SA for T2DM to adopt a healthy lifestyle and diet. Therefore, we have referred to the UK's guidelines on lifestyle as they share similarities in their healthcare systems.[19 20] These guidelines have been used to determine the most suitable behaviour targets for the Saudi context. We consulted healthcare professionals in the intervention setting, including a dietician, to establish these targets. Through collaborative discussions, we assessed and prioritised behaviours based on their potential impact, the feasibility of measurement, and achievability. As a result of this process, we have proposed four behaviour targets.[17]

The four general behaviour targets of the intervention were as follows[19 21]:
- ► Decrease carbohydrate intake for each meal.
- ► Use unsaturated fats as possible (avoid saturated fats).
- ► Do exercise for 30 min 5 days on a weekly basis.

► Monitor waist circumference and maintain it below (80 cm for women and 94 cm for men).

The aim was to gradually reduce carbohydrate intake while ensuring that participants met their nutritional requirements. The specific target amounts were communicated to participants during individualised dietary counselling sessions with health coaches, and participants were provided with guidance and meal plans to help them achieve and maintain the desired carbohydrate intake levels. The health coaching intervention used these targets as the benchmarks to track the progress and changes that the study population would make over 3 months.

We used diverse instruments, including questionnaires, interviews, focus groups and clinical measures, to assess the intervention's feasibility, acceptability and preliminary impact at two points: baseline and 3 months postintervention. In our mixed-methods study, purposive sampling was crucial to achieve data saturation and gather diverse perspectives. We conducted two distinct focus groups—one with six study participants and another with three health coaches—to assess the intervention's impact from multiple angles. Additionally, 14 semistructured interviews were carried out with selected participants to delve deeper into individual experiences and perceptions. These qualitative methods were complemented by quantitative evaluations at baseline and 3 months postintervention, using clinical measures such as HbA1c, body mass index (BMI), waist circumference and weight. The quantitative data were analysed using linear regression to determine the mean differences between the intervention and control groups, adjusted for baseline levels.

### Participant recruitment

Inclusion criteria were aged over 18, HbA1c level of 7.0% or higher, able to read and understand Arabic, having access to a personal mobile phone, being willing to complete the intervention period, be willing to stay in Riyadh (the capital city of SA) and be able to read, understand and sign the informed consent form.

The recruitment process involved advertising the intervention through multiple channels, including posters, brochures, social media and healthcare provider referrals. Interested and eligible individuals met the research team for a brief study overview and questionnaire completion, providing demographic data and baseline laboratory reports.

Participants were randomly assigned to one of two groups in a 1:1 ratio by an independent individual using a computer-generated random numbers system in Statistical Package for the Social Sciences (SPSS version number: 28.0.0.0), ensuring equal allocation chances. The two groups consisted of the intervention group, which received the health coaching intervention and the control group, which received standard care.

### Intervention delivery

The health coaching intervention was completed in 12 weeks. A detailed intervention curriculum was used in delivering the intervention activities towards achieving the four target behaviours (see online supplemental table 3).

Three health coaches, AA, AM and MA, who were qualified through the Saudi Ministry of Health, conducted the intervention sessions. They received training in biweekly face-to-face (F2F) meetings over 6 weeks on the intervention curriculum. A workbook was provided as a guide to help them deliver coaching sessions as planned. A detailed explanation has been published elsewhere.[15] It is important to note that participants did not directly receive or review the intervention curriculum. Our approach involved ensuring uniformity in intervention procedures by providing health coaches with the curriculum. Each coach was asked to document the BCTs they use in each session. Different BCTs were reported in each session. Each participant was assigned to one of three coaches at the baseline. Each participant received six sessions in total, which were delivered biweekly via video meetings and telephone coaching over 3 months. Due to the COVID-19 pandemic, the first and last meetings were conducted via video call instead of F2F as planned while the rest were via telephone. Participants were encouraged to contact their coaches via WhatsApp between sessions if they had any related concerns. 84 sessions were undertaken for all participants in the intervention group.

### Usual care (control group)

The participants in the control group were provided with the standard care, which involved receiving written materials and brochures on diabetes to increase awareness and promote the advantages of making changes to their health behaviours. Typically, individuals with T2DM have regular appointments with endocrine specialists to monitor their diabetes management. During these visits, the primary objective is to assess if any adjustments are necessary regarding the patient's medication, such as replacement, dosage increase or maintaining the current prescription.

### Data collection

Baseline data collection included demographic information, HbA1c, blood pressure, BMI, weight and waist circumference measurements. Subsequently, we asked participants to complete two electronic questionnaires: the Summary of Diabetes Self-Care Activities (SDSCA) scale and the Self-efficacy Scale for Diabetes. The postintervention data collection replicated the initial session, with repeat measurements of HbA1c, blood pressure, BMI, weight and waist circumference. Participants were also required to fill out the two initial questionnaires again, along with a Satisfaction Questionnaire designed specifically for the intervention group.

## Reach and retention

As per the study protocol,[15] this study aimed to recruit a minimum of 9% (n=30) of the required sample size for conducting a complete trial.[22] Each group was expected to have a minimum of 12 participants to ensure meaningful data interpretation for informing a future definitive RCT.[23] A total of 24 participants in each group would be required to detect a meaningful clinical difference in HbA1c. The retention rate was evaluated based on the predefined progression criteria described elsewhere,[15] with successful retention defined as achieving an 83% retention rate at the intervention endpoint.

## Fidelity assessment

All coaching sessions audio recorded and transcribed. An Arabic native speaker reviewed the entire transcription for accuracy. To ensure consistency and validity, a 10% sample of the transcription (approximately 50 min) was translated from Arabic to English, backtranslated and then examined by a peer reviewer. The data, including coaching sessions, time spent in each session, semistructured interviews, focus groups and questionnaires, were meticulously stored and managed for quality control. Specifically, all data were entered and stored in Microsoft Excel twice, independently by two research team members. This double-entry process served as a precautionary measure to minimise data entry errors and enhance the accuracy and reliability of the data collected throughout the trial. Adherence to the intervention was defined by the number of participants who completed five out of six coaching sessions, ≥84%.

## Feasibility

Evaluation of feasibility included participant eligibility and recruitment, retention, data collection adherence, and adherence to the intervention. We recorded participants' views about their engagement, capturing their reasons for interest, active participation and lack of interest. We tracked and logged the number and percentage of those who were eligible and interested, those who completed the intervention and those who withdrew to provide insight into the study's recruitment efficiency and participant retention.

## Acceptability

Acceptability of the intervention and its implementation was assessed using qualitative and quantitative data. The aim was to detect any potential challenges within the methodology used to deliver both the trial and the intervention. We conducted postintervention semistructured qualitative interviews and focus groups. The intervention's acceptability was primarily gauged through a Likert-scale Satisfaction Questionnaire (14 items) applied to the intervention group (n=14). This questionnaire, originally devised by the Dan Abrahams Healthy Living Centre and subsequently employed by DeJesus *et al* (2018)[24], incorporates a variety of question styles (open and closed). Using a Likert scale, it offered a response range from 0 (not at all) to 10 (very much) to capture participant feedback. Given that this scale had not been used in Arabic studies before, we translated it into Arabic and pretested it on a smaller subset before implementing it for the full study sample. In addition to quantitative measures, our mixed-methods approach involved semistructured interviews and focus groups to qualitatively assess the acceptability of the health coaching intervention. Thematic analysis was used to identify and interpret patterns within the qualitative data, aiming to understand the perceived benefits, challenges and overall acceptability of the intervention from the participants' perspective.

## Data analysis

We conducted both quantitative and qualitative analyses. For the quantitative part, descriptive analysis was undertaken including screening and recruitment rates, retention rates, adherence to coaching sessions and recruitment duration. We evaluated the use of BCTs and interactions between participants and coaches during these sessions. A Satisfaction Questionnaire captured participants' experiences while changes in diabetes self-management and patient self-efficacy were assessed at the beginning and end of the study. The data were entered into Excel and subsequently transferred to SPSS and Stata Statistical Software (STATA 17) for advanced analysis, which included a linear regression to estimate the mean differences in outcomes.

For the qualitative aspect of our study, purposive sampling was employed to ensure data saturation in conducting interviews and focus groups. We conducted a total of 14 semistructured interviews and two focus groups, engaging in detailed discussions with participants. The qualitative data were analysed using NVivo software through a reflexive thematic analysis process, adhering to six systematic phases: familiarising ourselves with the data, coding, generating themes, revising themes, defining and finally reporting.[25] Initial coding was conducted by the researcher (ANA), followed by a collaborative review with the research team. During this process, a consensus on the emerging themes was reached through regular team meetings and discussions, ensuring accuracy and comprehensiveness in the identification and categorisation of themes.

In addition, we integrated the qualitative and quantitative data for comprehensive understanding.[26] Using a joint display table (see online supplemental table 4),[27] we compared and combined these data types. The display helped us identify areas of agreement (convergence), disagreement (divergence) and enhancement (expansion) between the two data types.[28] In the integrated data, we denoted convergence with '=', complementarity with '+' and inconsistencies with '≠'.

Although this feasibility study may not conclusively determine intervention effects, it provides valuable insights for estimating the sample size for a larger trial based on the mean difference and SD of HbA1c.

**Table 1**  Summary of the participants' demographic characteristics

| Intervention group | | | | Control group | |
|---|---|---|---|---|---|
| | | Total N (%) | Median (IQR) | Total N (%) | Median (IQR) |
| Gender | Male | 6 (42.9) | | 7 (46.7) | |
| | Female | 8 (57.1) | | 8 (53.3) | |
| Age year | | 14 | 54.5 (44.5–59.25) | 15 | 54 (50–60) |
| Marital status | Married | 14 (100.0) | | 14 (93.3) | |
| | Single | 0 (0.0) | | 1 (6.7) | |
| Monthly income | Less than SR5000 | 4 (28.6) | | 7 (46.7) | |
| | SR5000–SR10 000 | 2 (14.3) | | 0 (0.0) | |
| | SR10 000–SR15 000 | 0 (0.0) | | 2 (13.3) | |
| | More than SR15 000 | 1 (7.1) | | 1 (6.7) | |
| | Prefer not to declare | 7 (50.0) | | 5 (33.3) | |
| Education level | Illiterate | 0 (0.0) | | 0 (0.0) | |
| | Primary school | 3 (21.4) | | 3 (20.0) | |
| | Secondary school | 2 (14.3) | | 2 (13.3) | |
| | High school | 3 (21.4) | | 4 (26.7) | |
| | Bachelor's degree | 2 (14.3) | | 3 (20.0) | |
| | Diploma | 1 (7.1) | | 0 (0.0) | |
| | Can read and write | 3 (21.4) | | 3 (20.0) | |
| Since when you were diagnosed with type 2 diabetes | Less than a year | 2 (14.3) | | 0 (0.0) | |
| | 1–3 | 6 (42.9) | | 0 (0.0) | |
| | 3–5 | 1 (7.1) | | 2 (13.3) | |
| | 5–7 | 0 (0.0) | | 1 (6.7) | |
| | 7–10 | 1 (7.1) | | 6 (40.0) | |
| | More than 10 years | 3 (21.4) | | 6 (40.0) | |
| | Do not know | 1 (7.1) | | 0 (0.0) | |
| Do you use diabetes medications? | Yes | 12 (85.7) | | 14 (93.3) | |
| | No | 2 (14.3) | | 1 (6.7) | |

## RESULTS
### Participant characteristics

The control group (mean age=53.40, SD=8.47) was slightly older than the intervention group (mean age=52, SD=8.32). The majority were married, with monthly incomes ranging from less than SR5000 (28.6%) to more than SR15 000 (7.1%); 50.0% chose not to declare. About a quarter had completed high school, and an equal proportion had primary education or were literate without formal education. Approximately 31% had lived with T2DM for over 10 years, with 24.1% of the intervention group having T2DM for 1–3 years. The majority (89.7%) were on diabetes medications. Table 1 summarises the demographic characteristics of both groups.

### Feasibility of the intervention
#### Eligibility and recruitment

The recruitment process took about 5 weeks, starting on 1 May 2021 to 5 June 2021. Leaflets were used for advertising the intervention and were distributed in different places in the hospital, including in waiting areas and on hospital wall notice boards. In addition, we met physicians in person to introduce the intervention and gave them leaflets with more details to encourage their patients to join in the study.

Of the 42 potential participants initially identified and assessed for eligibility, 38 met the study criteria, resulting in a screening rate of approximately 90%. These 42 individuals were referred by different sources. 22 were referred by their doctors while the others were recruited through various methods. Two were from leaflets, 8 were from suggestions by friends or relatives and 10 were directly engaged at the diabetes clinic. 12 were excluded, of which 4 were ineligible because they did not meet the intervention inclusion criteria for the following reasons: 1 had no access to a personal mobile phone/smartphone, 1 patient was diagnosed with T1DM, 1 had A1c below 7% and 1 was unable to do preassessments and postassessments. Eight were excluded for other reasons; two did not

respond and six for different reasons, for example, being too busy. 30 eligible patients were recruited to take part in the intervention. Out of the 38 eligible patients, 30 were successfully recruited and consented to participate in the intervention, yielding a recruitment rate of approximately 79%. All of them completed baseline assessments and questionnaires, and then randomly, 15 patients were allocated to the intervention group while the other 15 participants were assigned to the control group. One person who was part of the intervention group withdrew from the trial prior to the first session and was thus not included in the study (see online supplemental figure 1, CONSORT diagram for more details).

### Intervention delivery

The intervention started with the first session on 5 June 2021 and ended on 23 September 2021. The average time spent per participant was between ~17.7 and 25.5 in each session. The total range time of all sessions per patient was 109–153 min, with an average of 120.8 (SD=13.7). See online supplemental table 5 for more details about the time duration of each session.

### Retention of participants

30 eligible participants consented to participate in the trial and completed the baseline assessment randomly allocated either to the coaching group (n=15) or the control group (n=15). Of those, all 29 who started the intervention and took the first session remained till the endpoint and completed all the intervention activities. Only one participant allocated to the coaching group had withdrawn and discontinued the study before the first session (retention rate=97%). All 29 have completed the intervention endpoint assessments. The main progress criteria and feasibility measurements are summarised in online supplemental table 6.

### Adherence to the coaching sessions

15 out of 30 participants were randomly allocated to the coaching group. Before the first session, one participant withdrew (P12) due to family issues. The rest of the 14 have completed all their coaching sessions (adherence rate 100%), which met the predetermined progression criteria of adhering to ≥84%. Half of the participants took their sessions at the planned time as previously scheduled. However, health coaches have rescheduled different sessions for seven participants for different reasons. The rescheduled sessions varied between 1 and 2 sessions for each one of them. Out of 84 sessions, only 11 sessions have been rescheduled (13%). The reasons for the rescheduling were home/work/appointment conflicts (n=6 sessions), travel (n=1 session), illness (n=3 sessions) and being busy (n=1 session). All participants started their coaching sessions at the same time on 5 June 2021, except four participants who started later on 1 July 2021. A 15-day delay because their coach could not begin at that time due to family issues. In addition, the same four participants had another delay between the fourth

and fifth sessions due to the coach's college exams. This led them to finish their last session on 23 September 2021.

### Data collection adherence

Participants recruited for the study were invited to Al-Zulfi General Hospital on 10 June 2021. During this visit, they were given consent forms and information sheets about the intervention, providing them with an opportunity to ask questions and discuss any concerns related to their participation. Furthermore, baseline data, including measures of HbA1c, blood pressure, BMI, weight and waist circumference, were collected during this visit. Due to the restrictions imposed by COVID-19, all required measurements and paperwork were completed in a single visit. Participants were then asked to complete two electronic questionnaires, the SDSCA scale and the Self-efficacy Scale for Diabetes, during the hospital visit on 10 June 2021. The baseline data collection was successfully completed by all 30 participants, yielding a 100% completion rate. The time taken to complete the questionnaires ranged from 7 to 12 min, with no participants reporting difficulties or issues in the completion process.

At the end of the intervention, participants were once again invited to the hospital to gather postintervention data. This second data collection session was much like the first, where we again measured HbA1c, blood pressure, BMI, weight and waist circumference. Participants were also asked to complete the initial two questionnaires, with the addition of a Likert-scale Satisfaction Questionnaire for the intervention group only. This additional questionnaire resulted in a slightly longer completion time, ranging from 9 to 15 min, due to the inclusion of open-ended questions and certain items requiring justifications or explanations for selected responses. Overall, all participants effectively completed the data collection process, demonstrating successful adherence to our study's data collection procedures.

### Progression criteria

The feasibility measurements and predetermined progression criteria yielded positive results. The screening rate was 90%, exceeding the target of 80%, suggesting proceeding to the future definitive RCT. The recruitment rate reached 79%, demonstrating robust engagement with the target population. The retention rate at 3 months was 97%, surpassing the required 83% rate. Baseline data collection adherence and intervention adherence both achieved 100%. Endpoint data collection adherence was 97.6%, with only one participant withdrawing before the first session. Overall, these results support the decision to proceed to the future definitive RCT, indicating the feasibility and acceptability of the health coaching approach for individuals with T2DM in SA.

### Sample size

The sample size for this feasibility study was determined in consultation with a statistician and was guided by practical considerations as recommended by the CONSORT

guidelines for feasibility trials.[29] Our aim was not to test the intervention for statistical significance but rather to assess the feasibility and variability of outcomes to inform the planning of a definitive RCT. The choice of sample size was also influenced by the need to manage resources efficiently and to ensure a broad enough representation of the target population to capture initial insights into the intervention's acceptability and implementation.

Furthermore, the statistician advised that for the upcoming main trial, a sample size calculation should be based on a clinically significant mean difference of 0.5% and an SD of 1.4, given the substantial effect size of −0.93 observed in this feasibility study. With these parameters, using a power of 0.8 and an alpha level of 0.05, we estimated that 125 participants per group would be necessary. This sample size would not only allow us to detect a clinically significant difference but also accommodate subgroup analyses to explore variations in effectiveness across different settings or patient groups.

## Acceptability of the intervention

Acceptability and suitability of the intervention were assessed through a questionnaire (Likert-scale Satisfaction Questionnaire, 14 items) for the intervention group (n=14). Using a Likert scale of 0 (not at all) to 10, most participants' responses show that the intervention affected their behaviour 'quite a bit' (mean: 8.2 (SD=2.3)). Participants replied, 'very much' (mean 9.2 (SD=1.4)) when asked how much participation in the intervention helped them establish a personal vision of wellness. When asked how the intervention boosted their confidence in taking actions toward improved well-being, the overall response was 'very much' (mean 9.2 (SD=1.4)). Their motives for making efforts toward enhanced well-being were 'quite a bit' (mean 8.3 (SD=1)). The average response to the usage of goal setting was 'very much' (mean 9.2. (SD=0.9)). Whereas their mean response to their use of problem-solving abilities was 'quite a bit' (mean 8.6 (SD=1.4)). Participants shared that the intervention greatly assisted them in overcoming obstacles and achieving increased levels of wellness, with a high mean score of 9 (SD=1.1) reported. From the 1st–8th question, participants' responses are presented in online supplemental table 7.

Participants were asked three 'yes' or 'no' questions, and they all replied 'yes' when asked whether they expected to continue making improvements (n=14, 100%). The majority (n=13, 92.9%) responded with 'yes' the intervention met their expectations. All 14 participants who were asked whether they would recommend the intervention to others responded with a positive 'yes'.

## Thematic analysis findings

The thematic analysis of the qualitative interviews and focus groups underscored the intervention's positive reception, with emergent themes revealing enhanced self-efficacy, increased motivation and the supportive dynamics of the health coaching relationship, which

participants found to be a significant aspect of the intervention's acceptability. We analysed 1691 min of coaching sessions to identify the BCTs used (see online supplemental table 8) and transcribed 497 min of focus groups and interviews. A native Arabic speaker confirmed the accuracy of the entire transcription. To further ensure validity, about 10% of the transcription (50 min) underwent a back-translation process checked by a professional native speaker. Throughout the intervention, the first author (ANA), acting as the researcher, maintained field notes. Due to COVID-19 limitations, all interviews and focus groups were conducted online.

## Evaluation of participants and health coaches' experiences

Various qualitative methods were used to assess participant experiences during the intervention, including interviews, focus groups and field notes. After the intervention, two focus groups took place. Participants were given time to understand the questions before engaging in interviews and focus groups, fostering a deeper discussion. The researcher (ANA) conducted all focus groups and interviews, enabling active interaction and understanding of participant perspectives.

Between 15 June 2021 and 23 September 2021, 14 semistructured interviews were conducted with seven participants (4 males and 3 females) from the intervention group, both prestudy and post study. Participants predominantly had a high school education or less, and the median age was 59. Three had long-standing diabetes of over 10 years, three had a recent diagnosis of less than 3 years and one was uncertain of the duration. Six of the seven were on diabetes medication.

Two poststudy focus groups were conducted online. One focus group comprised six participants (three males and three females) from the intervention group with a median age of 52.5 years. The majority had a high school education or less, and most were on diabetes medication with a mix of recent and long-term diagnoses. The demographic characteristics of the participants can be seen in table 2.

Three female health coaches from King Fahad Medical City in Riyadh formed the second focus group to share their intervention delivery experiences. All had health coaching backgrounds, with two holding master's degrees in health education. The session lasted 265 min.

We used thematic analysis with inductive and deductive coding to evaluate the trial's acceptability and feasibility. Open coding initially identified emergent themes, with deductive coding following, grounded in qualitative research guidelines and the Medical Research Council framework.[30]

The analysis of translated transcripts produced four key themes, two of which were predetermined. The first theme pertained to trial design, conduct and processes, divided into four subthemes covering different aspects of intervention implementation. The second theme focused on intervention content and delivery, consisting of subthemes on intervention components, perceived

**Table 2** Participants' demographic characteristics

| Qualitative interview participants (n=7) | | | | Focus group participants (n=6) | |
|---|---|---|---|---|---|
| Demographic characteristics | | Number | % | Number | % |
| Gender | Male | 4 | 57.1 | 3 | 50 |
| | Female | 3 | 42.9 | 3 | 50 |
| Age | Years | 43–62 | Median 59 | 34–60 | Median 52.5 |
| Marital status | Married | 7 | 100 | 6 | 100 |
| Monthly income | Less than SR5000 | 1 | 14.3 | 2 | 33.3 |
| | SR5000–SR10 000 | 1 | 14.3 | 1 | 16.7 |
| | SR10 000–SR15 000 | 1 | 14.3 | 0 | |
| | Prefer not to declare | 4 | 57.1 | 3 | 50 |
| Education level | Primary school | 2 | 28.6 | 1 | 16.7 |
| | Secondary school | 2 | 28.6 | 1 | 16.7 |
| | High school | 2 | 28.6 | 3 | 50 |
| | Bachelor's degree | 1 | 14.3 | 1 | 16.7 |
| Since when you were diagnosed with type 2 diabetes | Less than a year | 2 | 28.6 | 1 | 16.7 |
| | 1–3 | 1 | 14.3 | 3 | 50 |
| | More than 10 years | 3 | 42.9 | 1 | 16.7 |
| | Do not know | 1 | 14.3 | 1 | 16.7 |
| Do you use diabetes medications? | Yes | 6 | 85.7 | 5 | 83.3 |
| | No | 1 | 14.3 | 1 | 16.7 |

consequences, feasibility and acceptability in practice, and reach and dose. The third theme, intervention mechanism, encompassed subthemes around COM-B model application, use of BCTs and intervention curriculum and intervention impact. The fourth theme, future RCT, captured participant and coach suggestions. The thematic map used in the analysis is outlined in online supplemental figure 2.

## THEME 1: INTERVENTION DESIGN, CONDUCT AND PROCESSES

This theme primarily focuses on the implementation and design of the intervention, along with the processes involved. Recruitment challenges during the pandemic prompted a shift to remote settings. This transition not only made recruitment a challenging task but also stretched the recruitment period beyond the planned time frame.

Notably, a health coach stated that the recruitment phase was complicated as clinics moved online., '…*the recruitment phase, it was complicated only because all clinics become online and we could not see people with type 2 diabetes come to the hospital as usual, so the chance of meeting those people was very rare.*' (Focus Group with Coaches, FGC01, Female).

Misunderstandings about health coaching in SA arose from a lack of understanding and familiarity. This highlighted the need for improved communication with recruiters, as one health coach observed, '…*from my*

*communication with recruiters, some of them did not understand, so participants may get a wrong idea*' (Focus Group with Coaches, FGC01, Female).

Limited hospital access due to public health guidelines complicated recruitment further. However, the suggestion to recruit from a secondary hospital made the recruitment process manageable. Participants and health coaches recommended physicians as intermediaries for patient recruitment due to the trust and rapport they have established with patients; as one of them stated, '…*I think one of the best and most reliable ways to reach and recruit patients is to start from their direct physicians by suggesting the program for patients and encourage them to join*' (Focus Group with Coaches, FGC02, Female).

Proposing clear communication about programme expectations aimed to reduce dropout rates, with one participant suggesting to '*ask participants to sign a contract to commit and if you are hesitating or you have the intention to withdraw to tell us [referred to program provider] at the beginning to give the opportunity to someone who needs it*' (Focus Group with Participants, P03, Female).

Health coaches and participants expressed general satisfaction with the intervention's design, and they believed that it effectively accommodated the local context of Saudi society. One participant expressed, '*I'm very very very satisfied and I think the program is easy to follow and very acceptable as it gradually improves different skills which last with a patient after the program…it is completely*

*different from other programs…'* (Interview with Participants, P13, Male).

Additional workload challenged health coaches, mixing intervention tasks with regular duties. The COVID-19 pandemic further complicated matters by requiring greater flexibility in scheduling coaching sessions, particularly for participants with children engaged in online learning. To alleviate these issues, health coaches proposed securing formal workplace approval to dedicate full time to the intervention. As one health coach noted, '*As you know and because of the workload, I was thinking of withdrawing from the intervention… I think it is imperative to work a full-time job as a health coach in the intervention, not as a volunteer, and get formal permission from my work on this…'* (Focus Group with Coaches, FGC01, Female).

## THEME 2: INTERVENTION CONTENT AND DELIVERY

This theme encompassed multiple subthemes reflecting on the intervention's implementation, including its acceptability and adaptability within the Saudi context, potential unintended consequences, the feasibility of the practical application and intervention reach and dosage.

Health coaches suggested modifications for better acceptability and suitability of the intervention in the Saudi context. Emphasising the need for professionalism and structure, they proposed using formal platforms for communication, as one coach stated, '*…communication with the participant should be through a formal platform or application… to make the program more formal and organised…'* (Focus Group with Coaches, C02, Female). They also suggested the importance of physical meetings or video calls for a more serious engagement, as expressed by another coach, '*I prefer to have a physical meeting where I can sit with the patient face-to-face…'* (Focus Group with Coaches, FGC01, Female).

Coaches highlighted the necessity of prepared private spaces for coaching sessions. to give a professional image to the intervention. As one coach expressed, '*…As a coach, there should be a certain prepared private place at work to conduct all of my coaching sessions…'* (Focus Group with Coaches, C02, Female).

Participants preferred visual coaching sessions over phone calls for effective communication. Coaches felt that seeing the participant was vital for effective communication and coaching. A health coach expressed this, saying, '*…visual interaction with participants is very very important, you know the importance of reading and understanding the body language for me as a coach…'* (Focus Group with Coaches, C03, Female).

Participants praised the intervention for its straightforwardness and effectiveness in helping them make progress towards their goals. Some suggested a desire for additional sessions to further adopt and maintain new habits. One participant commented, '*I think three months was enough'* (Interview with Participants, P03, Female), *while another remarked, '…it enabled me to stay committed,*

*continue, and achieve my goals'* (Interview with Participants, P09, Male).

Participants appreciated the use of health coaching techniques, such as action plans and SMART goals, which helped set achievable targets and keep them motivated. The flexibility and graduality of the programme were also well received, as one participant mentioned, '*…the most thing I liked about the program was the graduality to achieve goals…'* (Interview with Participants, P13, Male).

Coaches confirmed the intervention content's acceptability and observed its positive impacts. They also suggested the addition of monthly sessions or an extension of the intervention to 6 months for better patient progress monitoring and maintenance. A coach stated, '*… I suggest the intervention last longer… I think it would be good if the intervention lasts six months with a session each month …'* (Focus Group with Coaches, FGC01, Female).

## THEME 3: INTERVENTION MECHANISM

This theme emphasises the practical application and effects of the COM-B health coaching model. Coaches received specialised training on the curriculum and BCTs, using several tools for efficient delivery. Notably, coaches documented the BCTs employed in each session, enabling tailored strategies to target specific behaviours. Participants appreciated this approach, stating, '*The BCTs helped me during each session to address some behaviours…'* (Focus Group with Coaches, FGC01, Female).

The COM-B model played a crucial role in the intervention, targeting four behaviours. Participants noted marked improvements in their knowledge and motivation preintervention and postintervention. Testimonials showed that they were more knowledgeable about their condition, diet and physical activity. One participant said, '*I became more knowledgeable…now I know diet types and what carbohydrates mean…'* (Focus Group with Participants, P05, Female).

The intervention also impacted participants' social and physical environment, leading to a supportive atmosphere for behavioural change. Participants indicated that the changing social environment in SA encouraged the adoption of a healthier lifestyle and influenced their social networks. As one participant put it, '*While I was changing some of my old habits, my social network adapted…'* (Interview with Participants, P03, Female).

Health coaching provided a more personalised approach than traditional care, fostering motivation and leading to a change in participants' views about their condition. This motivation was further fuelled by the noticeable changes in outcomes such as HbA1c levels. Participants recognised the power of lifestyle changes and felt empowered by their improved health results, stating, '*…after the intervention, I realised modifying the lifestyle is the actual treatment…'* (Interview with Participants, P13, Male). Consequently, there were requests to extend the intervention, given the observable impacts, including reductions in A1c levels and body weight.

Participants in SA observed notable differences between the novel concept of health coaching and traditional care. The former was seen as superior, with participants expressing dissatisfaction with traditional care's impersonal nature. Health coaching, in contrast, provided a more personalised approach, allowing patients to voice their concerns. As one participant expressed, '*I would choose this program without hesitation; at least I finish each session while I'm comfortable…*' (Interview with Participants, P03, Female).

Traditional care was perceived as a top-down process, with patients passively receiving preprepared information or plans. The health coaching intervention, however, positioned participants as active stakeholders in their own care by encouraging goal-setting and plan creation. A participant noted, '*Your program helped me use new skills such as setting plans and having short-term and long-term goals…*' (Interview with Participants, P08, Male).

Frustration and even anger marked participants' feelings about traditional care, citing time constraints, a lack of empathy and the expense of private consultations. Health coaching was seen as a way to alleviate these issues, with participants suggesting the incorporation of health coaching clinics in hospitals to facilitate meaningful patient discussions and foster better disease self-management. A participant suggested, '*I think it is important to recruit a coach in the hospital… 90% of patients with chronic diseases need to sit with someone to have meaningful discussions…*' (Interview with Participants, P08, Male).

## THEME 4: IMPROVEMENT OPPORTUNITIES IN THE FUTURE HEALTH COACHING RCT

As the inaugural health coaching intervention within the Saudi healthcare system, this initiative stands as a landmark reflecting the strategic direction of the Saudi Ministry of Health towards more proactive, patient-centred care. In anticipation of scaling up such interventions, participants and health coaches provided constructive feedback for refinement. Participants expressed a preference for more frequent coaching interactions and the opportunity for extended F2F engagements, suggesting a deeper and more personal connection could enhance the intervention's impact. '*Having more time and direct sessions would greatly benefit the experience*' shared one participant (Interview with Participants, P01, Female).

Health coaches recommended the integration of multidisciplinary expertise, particularly dietitians and physicians, to provide comprehensive care and address the complex needs of diabetes management. '*Incorporating a nutritionist and a physician would offer a more rounded approach to patient care,*' proposed a coach (Focus Group with Coaches, C03, Female). This suggestion is indicative of a holistic strategy, recognising the multifactorial nature of diabetes.

To bolster participant engagement, the provision of incentives was suggested, alongside enhanced access to healthcare services. Additionally, health coaches proposed the creation of educational content, such as a succinct video delineating the health coaching concept, to facilitate a clearer understanding for future participants. '*A visual introduction to health coaching might bridge the initial knowledge gap,*' a coach highlighted (Focus Group with Coaches, C02, Female).

### Preliminary effects of the health coaching intervention

Our quantitative analysis, using a linear regression model adjusted for baseline levels, revealed significant health improvements from the health coaching intervention. Specifically, HbA1c levels in the intervention group decreased significantly, with a mean difference of −1.86 (95% CI −2.71 to −1.01, p<0.001) compared with the control group. Secondary outcomes further underscored the intervention's potential efficacy; BMI decreased by a mean of −1.02 (95% CI −2.01 to −0.041, p=0.042) and waist circumference reduced by −6.89 (95% CI −10.17 to −3.61, p<0.001). While changes in mean arterial pressure (MAP) and weight were observed, they did not reach statistical significance (MAP mean difference: −0.68, 95% CI −7.46 to 6.09, p=0.83; weight mean difference: −2.58, 95% CI −5.25 to 0.082, p=0.057). Detailed outcomes, including improvements in patients' self-efficacy and diabetes self-management, are delineated in tables 3 and 4.

### Patients' self-efficacy questionnaire

The self-efficacy of patients was assessed using an eight-item scale,[31] where scores ranged from 1 (least confident) to 10 (most confident). Higher mean scores indicated greater self-efficacy. Both intervention and control groups displayed an increased average response from baseline to the study endpoint, with higher improvements observed in the intervention group (see online supplemental table 9). The groups exhibited a statistically significant difference (p=0.006, 95% CI 0.713 to 3.839) in confidence related to controlling their condition and maintaining exercise. Similarly, a significant difference (p=0.006, 95% CI 0.82 to 4.35) was noted in confidence regarding appropriate food choices when hungry. This suggests the intervention group saw greater improvements in self-efficacy compared with the control group.

### Diabetes Self-Care Activity

The Diabetes Self-Care Activity scale,[32] scoring 0 (no adherence) to 7 (complete adherence), measured patients' adherence to specific activities over the past week. Average scores showed overall improvements in self-care for both the intervention and control groups across all activities, with the intervention group often meeting or exceeding American Diabetes Association recommendations (see online supplemental table 10).[33] For instance, adherence to a healthy eating plan in the intervention group increased from an average of 1.64–5.00 days per week. Physical exercise also increased in both groups, with a larger improvement in the intervention group (3.57–5.64 days). Similarly, there was improved

**Table 3** Groups means and SDs at study baseline and endpoint

|  | Group | N | Mean | SD |
|---|---|---|---|---|
| A1c level at the study baseline | Intervention group | 14 | 9.12 | 2.50 |
|  | Control group | 15 | 8.73 | 1.69 |
| A1c level at the study endpoint | Intervention group | 14 | 7.29 | 1.38 |
|  | Control group | 15 | 8.93 | 1.81 |
| Mean arterial pressure at the study baseline | Intervention group | 14 | 91.7 | 13.56 |
|  | Control group | 15 | 86.73 | 11.66 |
| Mean arterial pressure at the study endpoint | Intervention group | 14 | 92.57 | 9.11 |
|  | Control group | 15 | 90.33 | 12.91 |
| Weight in kg at the study baseline | Intervention group | 14 | 83.50 | 11.97 |
|  | Control group | 15 | 81.21 | 12.76 |
| Weight in kg at the study endpoint | Intervention group | 14 | 80.70 | 10.80 |
|  | Control group | 15 | 81.21 | 12.60 |
| BMI at the study baseline | Intervention group | 14 | 30.39 | 4.57 |
|  | Control group | 15 | 30.87 | 5.43 |
| BMI at the study endpoint | Intervention group | 14 | 29.39 | 4.33 |
|  | Control group | 15 | 30.86 | 5.318 |
| Waist circumference (cm) at the study baseline | Intervention group | 14 | 109.35 | 13.60 |
|  | Control group | 15 | 106.66 | 10.22 |
| Waist circumference (cm) at the study endpoint | Intervention group | 14 | 102.21 | 12.55 |
|  | Control group | 15 | 106.76 | 9.91 |

BMI, body mass index.

**Table 4** Outcomes mean differences taking into account the baseline value regression

| Endpoint HbA1c | Mean difference | t | P value | d | (95% CI) |
|---|---|---|---|---|---|
| Group* | −1.864463 | −4.50 | <0.001 | −0.93 | −2.716065 to 1.012862 |
| Baseline HbA1c | 0.566714 |  | <0.001 |  | 0.3601448 0.7732833 |
| Endpoint MAP | Mean difference | t | P value |  | (95% CI) |
| Group* | −0.6833264 | −0.21 | 0.837 |  | −7.463123 to 6.09647 |
| Baseline MAP | 0.5809645 | 4.38 | <0.001 |  | 0.3083653 to 0.8535638 |
| Endpoint weight | Mean difference | t | P value |  | (95% CI) |
| Group | −2.584974 | −1.99 | 0.057 |  | −5.252461 to 0.0825133 |
| Baseline weight T1 | 0.9090889 | 16.84 | <0.001 |  | 0.7981218 to 1.020056 |
| Endpoint BMI | Mean difference | t | P value |  | (95% CI) |
| Group | −1.024214 | −2.14 | 0.042 |  | −2.006754 to −0.0416741 |
| Baseline BMI T1 | 0.9322785 | 19.02 | <0.001 |  | 0.8315092 to 1.033048 |
| Endpoint WC | Mean difference | t | P value |  | (95% CI) |
| Group | −6.895251 | −4.32 | <0.001 |  | −10.17924 to 3.611259 |
| Baseline WC | 0.8732792 | 12.72 | <0.001 |  | 0.7322131 to 1.014345 |

*(Group=intervention mean–control mean).
BMI, body mass index; HbA1c, haemoglobin A1c; MAP, mean arterial pressure; WC, waist circumference.

management of carbohydrate intake in the intervention group, from 1.71 to 4.00 days per week.

## Integration results

The key findings from both quantitative and qualitative data, aligned with research objectives, were assembled in a joint display table (see online supplemental table 4). This table was structured to present the study's critical concepts and the corresponding progression criteria, which provide a comprehensive understanding of the mixed-methods findings. The phrase 'there is no significance in moving forward', as used within our progression criteria, draws from the principle of setting clear benchmarks for feasibility studies. These benchmarks are pivotal in assessing whether a study exhibits the potential for upscaling to a full-scale RCT. They embody a balance between the ambition to proceed and the pragmatism to address early issues, as recommended by best practices in trial management.[34] In our context, this phrase indicates that if participant recruitment, retention or adherence falls below the predetermined percentages, the study may not have the robustness required for expansion. Such a stance encourages rigorous scrutiny of recruitment rates and intervention adherence to ensure only well-substantiated research advances.

The integration of quantitative and qualitative data insights offered enhanced comprehension of findings collated and analysed separately. This approach bolstered the credibility and transparency of the overall results through data triangulation, allowing for comparison and validation of findings that a single method would not permit. The integration was critical to comprehensively understand the impacts of our intervention. Qualitative insights not only complemented the quantitative data but also provided deeper explanations for the patterns observed in numerical outcomes. For instance, while quantitative data showed significant improvements in HbA1c levels and BMI, qualitative findings offered nuanced insights into how and why participants made specific behavioural changes. Participants frequently cited the personalised feedback and motivational support from health coaches as pivotal, which helped them to adhere more consistently to their management plans. The integrated results substantiated the pertinence of mixed approaches, essential to the study's feasibility and acceptability exploration in the Saudi context for a large-scale intervention with patients with T2DM.

The study reported high screening, recruitment and retention rates. These results were corroborated by the integrated analysis, suggesting a promising potential for the study within the Saudi setting. This was further supported by participants' willingness to engage in the intervention, extend invitations to family members and their adherence to complete the intervention, despite some rescheduled sessions.

Notably, improvements were observed across five out of seven preliminary outcomes, which were validated by qualitative data. Participants and coaches reported positive experiences with the intervention's usability, design and content. Moreover, qualitative results echoed the high satisfaction levels among participants identified in the quantitative Likert-scale Satisfaction Questionnaire. Despite the challenge posed by session rescheduling flexibility for coaches, it facilitated participant adherence, as supported by quantitative data: all participants completed their sessions with only 11 sessions rescheduled.

## DISCUSSION

The findings from this feasibility study endorse advancing towards a larger RCT for a health coaching intervention among patients with T2DM in SA. Notably, the high rates of eligibility, recruitment and retention reflect the study's effectiveness in engaging participants, which is pivotal for ensuring the representativeness and reliability of the results.

Achieving a recruitment rate of 79% is particularly significant; it not only indicates the interest and willingness of the target population to participate but also reinforces the potential for this health coaching intervention to be scaled up and applied broadly within the Saudi healthcare setting. This recruitment performance, especially when contrasted with rates from prior studies, such as those by Basak Cinar and Schou and Cho et al,[35–37] highlights the tailored approach's resonance with the participants even amidst the operational challenges presented by the COVID-19 pandemic.

Retention is another key indicator of a study's appeal and the practicality of its intervention. The retention rate in this study exceeded that of a similar study (83.6%),[35] as well as the rates reported by Basak Cinar and Schou (87%)[37] and Cho et al (90%).[36] Our retention rate, which remained high throughout the study period, suggests that the intervention was well received and retained its relevance and value to the participants, motivating them to continue through the study's conclusion. The flexibility allowed in rescheduling coaching sessions contributed to this success, though it did increase the workload for the coaches, an aspect to consider in the planning of future RCTs.

As we reflect on the integrated quantitative and qualitative findings, we see confirmation of the feasibility of conducting a larger RCT, despite restrictions due to COVID-19. The comprehensive data collection, appropriate outcome assessments and participant adherence collectively point to the potential for effective execution and data gathering in a subsequent, more extensive RCT. Enhancing reach and diversity through a variety of recruitment methods, and the use of a concise explanatory video to describe the intervention, are strategies that can further streamline the process and solidify understanding among future participants.

The feasibility study used the COM-B model and the BCTs taxonomy, which proved to be effective in guiding intervention content and addressing participant barriers. The most frequently used BCTs aligned with previous

literature. The study adapted to telephone-based sessions due to COVID-19 restrictions, but a mixed approach with F2F interactions was preferred by both participants and coaches. A longer intervention duration of up to 6 months with more F2F sessions was recommended to ensure sustained behaviour changes.

The intervention's positive outcomes, particularly the reduction in HbA1c levels, align with evidence suggesting that interventions emphasising HbA1c as the primary outcome are more effective. This underscores the importance of selecting outcomes aligned with the intervention's central goal. This study supports previous findings,[38] indicating that using HbA1c as the primary intervention outcome can enhance effectiveness. Emphasising glycaemic control as the primary outcome significantly impacted participants' diabetes self-management in this study. Participants actively developed and used key skills to manage their condition, indicating the efficacy of the coaching model. Their satisfaction was notably linked to a meaningful reduction in HbA1c levels after the intervention, emphasising the holistic impact of the health coaching model.

Although feasibility studies are not formally powered to detect conclusive effects on interventions,[39] this study demonstrated the high acceptability and usability of the intervention in the Saudi context. However, concerns were raised by health coaches regarding their volunteer roles and the challenges of finding suitable locations for coaching sessions. These issues need to be addressed in planning for the larger RCT. The ongoing societal changes in SA, along with participants' satisfaction with the intervention, suggest potential acceptance for a large-scale RCT.

For the upcoming definitive RCT, it is crucial to integrate participant and coach feedback to refine the approach. Recommendations include lengthening the programme to 6 months with increased in-person interactions, involving multidisciplinary experts, regular coach coordination, participant incentives and a brief explanatory video to introduce health coaching.

This study introduces a patient-centred health coaching approach to SA, showcasing its effectiveness in managing T2DM and its potential applicability to other chronic conditions. The use of a detailed curriculum and the COM-B model has proven beneficial, supporting the Saudi Ministry of Health's move towards behaviour-based strategies and paving the way for broader implementation.

The study's positive reception and outcomes reinforce the viability of health coaching in T2DM management, offering insights for future research and substantiating its growing acceptance and practicality.

### Strengths and limitations of this study

This study applied the BCW framework, proving beneficial for a comprehensive understanding of the issue of T2DM in the context of SA through an in-depth analysis of prior literature. Accurate diagnosis of the identified problem, considering potential barriers and facilitators,

plays a pivotal role in the systematic roadmap. This process initiates with recognising the actual issue and concludes with specifying target behaviours necessary to achieve the intended goals. Employing the COM-B model, we conducted a narrative review to pinpoint facilitators and barriers in controlling T2DM. The unique approach involved employing the BCW, COM-B model and BCTs taxonomy, enabling a meticulous analysis of each health coaching session and the coding of each applied BCT. This resulted in a clear and detailed description of the intervention.

The inclusion of progression criteria, as recommended by the CONSORT 2010 statement, aided in decision-making for the larger trial.[40] The study findings, including the effect size and retention rate, will inform the estimation of the sample size for the future main trial. The study also used both the Summary of Diabetes Self-Care Activity questionnaire and the Self-efficacy Scale for Diabetes scale, providing a comprehensive assessment of health coaching's impact. A mixed-methods approach was employed to gather diverse perspectives, enhance validity and reliability, and produce more comprehensive findings.

However, the study is not without limitations, particularly concerning generalisability due to the specific setting and population. The small sample size may have led to potential type 2 errors, limiting the study's ability to detect certain effects. While the recruitment of the intervention sample from a public hospital may not be directly applicable to primary care settings, it could impact the broader generalisability of findings to other individuals.

It is essential to note that although statistically significant effects were detected, the study's acknowledgement of limitations in statistical power reflects an awareness of potential constraints in identifying smaller yet existing effects. These limitations should be considered when interpreting the study's outcomes.

### CONCLUSION

This mixed-methods feasibility RCT demonstrated the feasibility and acceptability of implementing a health coaching approach for individuals with T2DM in SA. The integration of qualitative and quantitative findings yielded positive outcomes, providing strong support for conducting a larger RCT to further investigate the effectiveness of health coaching in enhancing diabetes self-management among individuals with T2DM in SA. Importantly, this study offers valuable insights for future trials, particularly in terms of incorporating the COM-B model and BCTs taxonomy into the health coaching approach. The positive feedback received for implementing a patient-centred approach indicates that it could help manage other chronic diseases. This aligns with the Saudi Arabian Ministry of Health's behaviour-focused strategies, providing more evidence to support the expansion of health coaching. As a result, these insights have significant implications for the current management

practices of T2DM and beyond. Future research should also explore the social and environmental impacts of Vision 2030, the national vision of SA, on diabetes self-management. By considering contextual factors that may influence the implementation and outcomes of health coaching interventions for diabetes management in SA, further investigations will deepen our understanding and inform strategies to improve diabetes care and other chronic diseases in the country.

**Acknowledgements** We express our sincere gratitude to the study participants for their valuable time and willingness to participate. We are also grateful to KFMC and Al-Zulfi General Hospital for granting permission to conduct the intervention. Furthermore, we extend our thanks to the University of Sheffield Institutional Open Access Fund for their support towards this research. Finally, we would like to acknowledge that this paper will be published under a Creative Commons Attribution (CC BY) licence.

**Contributors** ANA was responsible for the study's conception and design, data acquisition, analysis, and interpretation, as well as the drafting of the manuscript, and acts as the guarantor of the work. ANA accepts full responsibility for the work and the conduct of the study, had access to the data, and controlled the decision to publish. EG contributed to the critical revision of the manuscript based on data analysis. AA, AM, MA, and ASA were responsible for the implementation of the intervention. All authors have thoroughly reviewed and approved the final version of the manuscript for publication.

**Funding** The authors have not declared a specific grant for this research from any funding agency in the public, commercial or not-for-profit sectors.

**Competing interests** None declared.

**Patient and public involvement** Patients and/or the public were not involved in the design, or conduct, or reporting, or dissemination plans of this research.

**Patient consent for publication** Not applicable.

**Ethics approval** This study involves human participants and ethical approval for this study was granted by both the University of Sheffield and the Institutional Review Board (IRB) committee at King Fahad Medical City (KFMC), with the assigned IRB log number: 21-062E. Participants gave informed consent to participate in the study before taking part.

**Provenance and peer review** Not commissioned; externally peer reviewed.

**Data availability statement** Data are available on reasonable request. Data supporting the findings of this study are available from the corresponding author on reasonable request. Access requests will be reviewed to ensure compliance with ethical and privacy guidelines. Please contact (aalmulhim@seu.edu.sa) for inquiries regarding data access.

**ORCID iD**
Abdullah N Almulhim http://orcid.org/0000-0003-1745-7528

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
