## [Reviewer comments · BMJ Open]

ARTICLE DETAILS

TITLE (PROVISIONAL)	Feasibility and Acceptability of a Tailored Health Coaching Intervention to Improve Type 2 Diabetes Self-Management in Saudi Arabia: A Mixed-Methods Randomised Feasibility Trial
AUTHORS	Almulhim, Abdullah; Alhowaish, Atheer; Madani, Alaa; AlQaddan, Munirah; Altuwalah, Abdulaziz S.; Goyder, Elizabeth

VERSION 1 – REVIEW

REVIEWER	Standley, Krys University of Montana, Rural Institute
REVIEW RETURNED	29-Aug-2023

GENERAL COMMENTS	This mixed-methods study provides valuable insights into health coaching that support the validity of the results obtained from both analytic methods. However, as it is currently written, there are numerous minor revisions related to clarity, accuracy, consistency, and replicability that need to be addressed for this manuscript to be suitable for publication, which are detailed in the following sections. Throughout the paper: Please define all acronyms immediately prior to their first use (e.g., BCTTv1, BCW, F2F, MRC, FGC, C01, etc.). Please proofread thoroughly for grammatical typos and proper punctuation. Examples include the sentences beginning “An RCT design to both...”, “Fifteen days of delay...” “The retention rate atthree3...”, “Despite the challenge posed by session rescheduling flexibility for coaches...”, and the phrase “behaviours barriers,” etc. Please also ensure that all headings have correct punctuation and capitalization and that in-text citations superscripts are placed consistently outside of punctuation marks. I suggest shifting from passive voice to active voice throughout the paper. This is especially salient in the thematic analysis, where statements such as “Challenges were encountered” fail to convey by whom the challenges were experienced. Introduction: On Page 3, you refer to obesity as a lifestyle factor. This could be better conceptualized as a disease symptom of other lifestyle factors. Materials and Methods: On Page 4, the study is described as “using a double-blind...” which appears to be inaccurate. In a double-blind study, neither participants nor researchers know which conditions the participants have been assigned to. Your use of blinding in this study was limited to the random assignment to conditions (intervention vs. control). Therefore, participants knew which group they were in and this study does not qualify as being a double-blinded study. Please omit this language from the manuscript.
--

	It is unclear in what ways the Behavior Change Wheel was incorporated into the intervention. Please add information to clarify this point. On Page 5, under Reach and retention, you state that “The study aimed to recruit a minimum of 9% (n = 30) of the required sample size for conducting a complete trial. I didn’t understand what this meant – it needs clarification in terms of what this means and what its purpose is. In the design section, you indicate the target population as “those with poorly managed T2DM”. A more precise statement would indicate that the population of interest is “adults with poorly managed T2DM.” The first of four general behavior targets of this intervention was “decrease carbohydrate intake for each meal”. How was this operationalized within the study? What quantity of carbohydrates was the target amount? The curriculum protocols are well-described and you state that coaches received a workbook with the curriculum protocols. However, more information should be provided about how the curriculum was presented to participants. Did participants receive or view any curriculum materials? In either case, it would be helpful to clarify these procedures. Were the health coaches for the study novice health coaches or were they certified health coaches? If they were previously certified, please describe their previous training. For readers outside of Saudi Arabia, it may be helpful to describe that Riyadh is the capital city of SA. Under “Fidelity assessment” what does it mean that “the data was stored twice in Microsoft Excel”? Please also include the purpose of this step. Results: Under participant characteristics, be more specific about the varying monthly incomes of the participants. Provide a range or some other tangible quantity, similar to your approach to the other characteristics you describe in this section. In Table 1, monthly income should be accompanied by a named monetary unit. In Table 3, participant height is described. However, height is not described elsewhere in the manuscript. Given that this information is captured in BMI, I suggest removing height from this table. I am unfamiliar with the label P>t used in Table 4. I suggest using the label p-value or significance value. Under “Intervention delivery,” I do not see a need to include the total of 1691 minutes of health coaching. However, you could add the total number of participants in the intervention condition to the final sentence in that paragraph, which discusses the total range of time per patient. I recommend that the content of the second paragraph under “Intervention delivery” should be made into a table. This table could replace Supplemental Table 5, which contains raw data that is unnecessarily detailed. Under “Data collection adherence,” please clarify if the two electronic questionnaires were also completed during the hospital visit on June 10, 2021. Under “Acceptability of the intervention,” I am presuming the numbers in parentheses after each mean value are the standard deviations. Please make this explicit. Please align the language of “observational notes” (under “Thematic Analysis findings”) and “field notes” (under “Evaluation of Participants and Health Coaches’ Experiences”) to clarify that
--	--

	these both refer to the same thing, if indeed this is the case. If not, please clarify. In this paper, you refer several times to “the researcher.” Although it is implied in the “Author contributions” that this is the first author, it would be helpful to clarify this in the text. I recommend altering the introductory language in the third and fourth paragraphs under “Evaluation of Participants and Health Coaches’ Experiences” from “first” and “second” focus groups to “one focus group with participants” and “one focus group with coaches” or something to that effect. As it is, the language sets the reader up to think there were two sets of focus groups with participants, which creates unnecessary confusion. Thematic analysis findings I am presuming that “P01,” “P08,” etc. indicate participant numbers. I suggest that you consider assigning pseudonyms and providing brief demographic information with the first mention of each participant who is cited, to contextualize the statements. It appears the acronym “FGC” indicates statements made by health coaches. This acronym needs to be either defined or replaced with a description of the coach. Is there more to say to qualify the statement that “This is Saudi Arabia’s first health coaching intervention”? Preliminary effects of the intervention Throughout this paper, the authors’ conceptualization of the lack of statistical power is in need of correction. The study may, indeed, have been underpowered to detect some effects that were present, resulting in potential Type 2 errors. However, if you were able to detect a statistically significant effect, then clearly you had the statistical power to do so. By updating your understanding of this concept, you can then adjust the ways in which you discuss your findings throughout this manuscript. The authors refer to effect sizes in the narrative but, as far as I can see, effect sizes were not reported and are notably absent from the results section and its related tables – they should be included alongside the reported p-values. This is a lengthy paper, even when including only the most relevant information. Therefore, it is important that the authors refrain from including superfluous content. Across all measurements (HbA1C, MAP, Weight, etc.) the average values for measurements of all participants at the beginning of the study are useful to convey. However, overall participant values of these measurements at the end of the study should be omitted, as the group differences at the study’s endpoint better describe participants’ trajectories. Similarly, mean differences for all participants should be omitted. I would suggest that all mentions of “significant reduction” or “significant difference” should be revised to explicitly indicate that these are “statistically significant” reductions or differences, to distinguish between statistical significance and practical importance. Integration Results: In Supplemental Table 4, the contents of the column titled “Predetermined progression criteria,” the meaning of the phrase, “there is no significance in moving forward” should be elaborated on. However, this should be done in the body of the text rather than in the table. Also, the content of the table should be condensed and repetition of phrases should be avoided. Discussion
--	---

	Review for conceptual clarity regarding the importance of recruitment rates, including the reference to recruiting a certain percentage of a study's sample. You state that "using HbA1C as the main intervention outcome can improve effectiveness." Please clarify your intended meaning of this statement by explaining how you conceptualize this process. Strengths and Limitations You state that "The study findings, including effect size and retention rate, informed the future main trial's sample size estimation." It is strange to include a past tense verb for a future trial, and this language makes what you are trying to say unclear. The first two sentences of the "Strengths and limitations" section are unclear in meaning. In what ways did the Behavior Change Wheel prove beneficial in understanding T2DM in SA through a comprehensive literature review? How did the COM-B model facilitate a narrative review? Please see my previous comments about shifting your conceptualization of the lack of statistical power. While it may be true that lack of statistical power limited the detection of statistically significant differences, you reported detecting these differences in many of the variables that were measured. Please check the last sentence for accuracy. Your statement about "the age cohort of older adolescents may limit applicability to younger individuals," yet the mean participant age you reported in this study was over 50 years of age. Conclusion Please clarify if the Ministry of Health is a Saudi Arabian institution or if it is based elsewhere. References: Many citations are missing the year published. Please include this information and ensure that all citations are complete. Please also ensure appropriate capitalization in references cited (e.g., reference #14). Citation formatting is inconsistent and should be corrected to be consistent. Tables and Supplemental Tables All acronyms used in tables (including table headings and within cells) should be defined in or under each table for convenient reference by readers.
--	---

REVIEWER	Chaudhry, Zoobia Wazir Johns Hopkins University
REVIEW RETURNED	05-Sep-2023

GENERAL COMMENTS	On page 5, it is not clear why researchers adopted measuring waist circumference as one of the behavior targets. I am not able to find it as one of goal self monitoring technique in cited reference (PMID: 29443421).
---

REVIEWER	Blatch-Jones, Amanda National Institute for Health Research Evaluation, Trials and Studies Coordinating Centre (NETSCC), University of Southampton
REVIEW RETURNED	23-Oct-2023

GENERAL COMMENTS	A couple of comments that could help to improve the manuscript and its purpose: The abstract and title could make it clearer that this is a feasibility study. The introduction is well written and contains sufficient evidence and previous literature to demonstrate the need for the research and its intended contribution to the existing literature. Under methods, what was the computer-generated random numbers system that was used? Under feasibility section, typo we recorded not we recorder The methods section does not explicitly state the intended qualitative aspect of the study apart from data analysis. How were they approached, was data saturation considered or was it purposive sampling to conduct the interviews and focus groups. How many participants were required, how many focus groups were intended with how many participants. How was the qualitative aspect of the study analysed, using what software or approach? How was consensus reached during coding the categorisation? A sub section specifically detailing the qualitative aspect of the study would strengthen the paper, as it is described as mixed methods and much attention is on the feasibility aspect of the study. The results of the qualitative aspect of the study are well written and contain relevant and important considerations that complement the feasibility study. Overall, this is a well written manuscript with minimal revisions required. The conclusion is backed up from the studies findings as well as current literature, whilst also detailing the studies limitations.
--

VERSION 1 – AUTHOR RESPONSE

Reviewers' comment	Our response
Throughout the paper:  1. Please define all acronyms immediately prior to their first use (e.g., BCTTv1, BCW, F2F, MRC, FGC, C01, etc.). 2. Please proofread thoroughly for grammatical typos and proper punctuation. Examples include the sentences beginning “An RCT design to both...”, “Fifteen days of delay...” “The retention rate aththree3...”, “Despite the challenge posed by session rescheduling flexibility for coaches...”, and the phrase “behaviours barriers,” 	 1- Thank you for pointing this out. We have now defined all acronyms immediately before their first use in the manuscript to enhance clarity and comprehension for readers. 2- Thank you for your feedback. We have carefully proofread the manuscript, corrected the mentioned grammatical typos and punctuation errors, and ensured consistent heading punctuation and capitalization. In-text citations with superscripts are now consistently placed outside punctuation marks. These revisions have improved the manuscript's quality. 3- Thank you for your valuable feedback. We have revised the paper to shift from passive voice to active voice throughout, particularly in the thematic analysis section.

etc. Please also ensure that all headings have correct punctuation and capitalization and that in-text citations superscripts are placed consistently outside of punctuation marks. 3. I suggest shifting from passive voice to active voice throughout the paper. This is especially salient in the thematic analysis, where statements such as “Challenges were encountered” fail to convey by whom the challenges were experienced.	
Introduction: On Page 3, you refer to obesity as a lifestyle factor. This could be better conceptualized as a disease symptom of other lifestyle factors.	The section on Page 3 has been updated to describe obesity more appropriately as a symptom stemming from lifestyle factors.
Materials and Methods: 1. On Page 4, the study is described as “using a double-blind...” which appears to be inaccurate. In a double-blind study, neither participants nor researchers know which conditions the participants have been assigned to. Your use of blinding in this study was limited to the random assignment to conditions (intervention vs. control). Therefore, participants knew which group they were in and this study does not qualify as being a double-blinded study. Please omit this language from the manuscript. 2. It is unclear in what ways the Behavior Change Wheel was incorporated into the intervention. Please add information to clarify this point. 3. On Page 5, under Reach and retention, you state that “The study aimed to recruit a minimum of 9% (n = 30) of the required sample size for conducting a complete trial. I didn’t understand what this meant – it needs clarification in terms of what this means and what its purpose is. 4. In the design section, you indicate the target population as “those with poorly managed T2DM”. A more precise statement would indicate that the	1- The term 'double-blind' has been omitted, and we appreciate your guidance in ensuring the accuracy of our study description. 2- Thank you for your insightful comment. To clarify the incorporation of the Behaviour Change Wheel (BCW) in our intervention, we have revised the 'Materials and Methods' section. The revision includes detailed explanations of how the BCW was utilized for intervention function identification, mapping of Behaviour Change Techniques, and conducting a behavioural analysis in line with the Theoretical Domains Framework. Additionally, we have noted that further details on the intervention are available in our previously published study protocol 3- We've revised the section for clarity 4- The text now accurately specifies 'adults with poorly managed T2DM' as the target population, aligning with the study's focus. 5- Clarified in the manuscript—targets and guidance on carbohydrate reduction were provided in personalized counseling sessions with health coaches. 6- In the revised text, we clarify that participants did not receive the curriculum directly; instead, health coaches were provided with the curriculum to ensure standardized intervention delivery. 7- Revised to specify: The intervention was delivered by three health coaches certified by the Saudi Ministry of Health. 8- Revised to define Riyadh as the capital city of Saudi Arabia in the inclusion criteria section.

population of interest is “adults with poorly managed T2DM.”  5. The first of four general behavior targets of this intervention was “decrease carbohydrate intake for each meal”. How was this operationalized within the study? What quantity of carbohydrates was the target amount? 6. The curriculum protocols are well-described and you state that coaches received a workbook with the curriculum protocols. However, more information should be provided about how the curriculum was presented to participants. Did participants receive or view any curriculum materials? In either case, it would be helpful to clarify these procedures. 7. Were the health coaches for the study novice health coaches or were they certified health coaches? If they were previously certified, please describe their previous training. 8. For readers outside of Saudi Arabia, it may be helpful to describe that Riyadh is the capital city of SA. 9. Under “Fidelity assessment” what does it mean that “the data was stored twice in Microsoft Excel”? Please also include the purpose of this step. 	 9- Revised to explain that the double-entry of data into Microsoft Excel by two independent team members was a quality control measure to reduce errors and increase data accuracy and reliability throughout the trial.
Results:  1. Under participant characteristics, be more specific about the varying monthly incomes of the participants. Provide a range or some other tangible quantity, similar to your approach to the other characteristics you describe in this section. 2. In Table 1, monthly income should be accompanied by a named monetary unit. 3. In Table 3, participant height is described. However, height is not described elsewhere in the manuscript. Given that this information is captured in 	 1- The section on participant characteristics has been updated to include a specific income range, demonstrating the diversity of the study population's economic backgrounds. 2- The monetary unit has been added to Table 1 to clarify the income range. 3- Height has been removed from Table 3 to avoid redundancy, as BMI already encapsulates this information. 4- The label in Table 4 has been updated to "p-value" for clarity 5- Intervention delivery section has been edited as suggested in comment number 6 6- The 'Intervention delivery' section has been revised as recommended 7- The sentence under 'Data collection adherence' has been updated to confirm that both electronic questionnaires were indeed completed during the hospital visit on June 10th, 2021

BMI, I suggest removing height from this table. 4. I am unfamiliar with the label P>t used in Table 4. I suggest using the label p-value or significance value. 5. Under “Intervention delivery,” I do not see a need to include the total of 1691 minutes of health coaching. However, you could add the total number of participants in the intervention condition to the final sentence in that paragraph, which discusses the total range of time per patient 6. I recommend that the content of the second paragraph under “Intervention delivery” should be made into a table. This table could replace Supplemental Table 5, which contains raw data that is unnecessarily detailed. 7. Under “Data collection adherence,” please clarify if the two electronic questionnaires were also completed during the hospital visit on June 10, 2021. 8. Under “Acceptability of the intervention,” I am presuming the numbers in parentheses after each mean value are the standard deviations. Please make this explicit. 9. Please align the language of “observational notes” (under “Thematic Analysis findings”) and “field notes” (under “Evaluation of Participants and Health Coaches’ Experiences”) to clarify that these both refer to the same thing, if indeed this is the case. If not, please clarify. 10. In this paper, you refer several times to “the researcher.” Although it is implied in the “Author contributions” that this is the first author, it would be helpful to clarify this in the text. 11. I recommend altering the introductory language in the third and fourth	8- The section has been edited for clarity: numbers in parentheses following mean values now explicitly denote standard deviations. 9- The manuscript has been revised to consistently use one term to eliminate confusion and ensure clarity regarding the notes taken during the study. 10- The manuscript has been updated to specify that 'the researcher' refers to the first author (AA), who conducted all focus groups and interviews, and maintained field notes throughout the study. 11- The section has been revised to clearly differentiate between the participant focus group and the health coach focus group to avoid confusion.
---	--

paragraphs under “Evaluation of Participants and Health Coaches’ Experiences” from “first” and “second” focus groups to “one focus group with participants” and “one focus group with coaches” or something to that effect. As it is, the language sets the reader up to think there were two sets of focus groups with participants, which creates unnecessary confusion.	
Thematic analysis findings  1. I am presuming that “P01,” “P08,” etc. indicate participant numbers. I suggest that you consider assigning pseudonyms and providing brief demographic information with the first mention of each participant who is cited, to contextualize the statements. 2. It appears the acronym “FGC” indicates statements made by health coaches. This acronym needs to be either defined or replaced with a description of the coach. 3. Is there more to say to qualify the statement that “This is Saudi Arabia’s first health coaching intervention”? 	 1- Thank you for the suggestion to add more context to the participant references. To provide clarity without redundancy, we have included detailed demographic information for each participant in Table 2. This table accompanies their first mention in the text. 2- The acronym 'FGC' has been clarified in the manuscript to explicitly denote 'Focus Group with Coaches'. 3- We have added contextual information and references to reinforce the statement.
Preliminary effects of the intervention  1. Throughout this paper, the authors’ conceptualization of the lack of statistical power is in need of correction. The study may, indeed, have been underpowered to detect some effects that were present, resulting in potential Type 2 errors. However, if you were able to detect a statistically significant effect, then clearly you had the statistical power to do so. By updating your understanding of this concept, you can then adjust the ways in which you discuss your findings throughout this manuscript 2. The authors refer to effect sizes in the narrative but, as far as I can see, effect sizes were not reported and are notably absent from the results section and its 	 1. Thank you for your valuable feedback. We appreciate your insights into the conceptualization of statistical power. We have carefully revised the manuscript to provide a more accurate and nuanced discussion of the study's findings. 2. As outlined in our previously published study protocol, our feasibility study aimed to assess recruitment and retention rates, and explicitly, to estimate the effect size, focusing on the primary outcome HbA1c. The effect size was reported under the sample size section and alongside the A1c p-value in the results table. 3- Thank you for your feedback. We appreciate your suggestion to streamline the presentation

related tables – they should be included alongside the reported p-values. 3. This is a lengthy paper, even when including only the most relevant information. Therefore, it is important that the authors refrain from including superfluous content. Across all measurements (HbA1C, MAP, Weight, etc.) the average values for measurements of all participants at the beginning of the study are useful to convey. However, overall participant values of these measurements at the end of the study should be omitted, as the group differences at the study's endpoint better describe participants' trajectories. Similarly, mean differences for all participants should be omitted. 4. I would suggest that all mentions of "significant reduction" or "significant difference" should be revised to explicitly indicate that these are "statistically significant" reductions or differences, to distinguish between statistical significance and practical importance.	of results. We have revised the manuscript accordingly. 4- Thank you for your suggestion. We have made the necessary revisions in the manuscript to explicitly state "statistically significant differences", as per your recommendation.
Integration Results: In Supplemental Table 4, the contents of the column titled "Predetermined progression criteria," the meaning of the phrase, "there is no significance in moving forward" should be elaborated on. However, this should be done in the body of the text rather than in the table. Also, the content of the table should be condensed and repetition of phrases should be avoided.	Thank you. The meaning of the phrase, "there is no significance in moving forward," has been elaborated on in the body of the text. Additionally, the content of Supplemental Table 4 has been condensed, and repetition of phrases has been addressed.
Discussion 1. Review for conceptual clarity regarding the importance of recruitment rates, including the reference to recruiting a certain percentage of a study's sample. 2. You state that "using HbA1C as the main intervention outcome can improve effectiveness." Please clarify your intended meaning of this statement by explaining how you conceptualize this process.	1- We have reviewed the discussion section for conceptual clarity regarding the importance of recruitment rates, including the reference to recruiting a certain percentage of the study's sample. Clarifications and improvements have been made to enhance conceptual clarity in this regard. 2- The manuscript has been revised to clarify that focusing on HbA1c levels as the primary outcome aligns with evidence indicating greater intervention effectiveness. This is further supported by our study's findings, where emphasizing glycemic control as the primary outcome

	led to significant self-management improvements by participants.
Strengths and Limitations  1. You state that “The study findings, including effect size and retention rate, informed the future main trial's sample size estimation.” It is strange to include a past tense verb for a future trial, and this language makes what you are trying to say unclear. 2. The first two sentences of the “Strengths and limitations” section are unclear in meaning. In what ways did the Behavior Change Wheel prove beneficial in understanding T2DM in SA through a comprehensive literature review? How did the COM-B model facilitate a narrative review? 3. Please see my previous comments about shifting your conceptualization of the lack of statistical power. While it may be true that lack of statistical power limited the detection of statistically significant differences, you reported detecting these differences in many of the variables that were measured. 4. Please check the last sentence for accuracy. Your statement about “the age cohort of older adolescents may limit applicability to younger individuals,” yet the mean participant age you reported in this study was over 50 years of age. 	 1. The manuscript has been updated to clarify that the effect size and retention rate obtained from this study will guide the calculation of the sample size for the upcoming main trial. 2. Revised to clarify the use of the BCW framework for literature analysis and the COM-B model for identifying T2DM management barriers and facilitators in the intervention design. 3- Revised the "Strengths and limitations" section to address this point: while the study did detect significant differences in several variables, the small sample size may still constrain the detection of differences in others, potentially affecting the generalizability of these findings. 4- The last sentence has been edited for accuracy.
Conclusion Please clarify if the Ministry of Health is a Saudi Arabian institution or if it is based elsewhere.	Revised to specify that the Ministry of Health's strategies mentioned in the study are indeed those of the Saudi Arabian Ministry of Health.
References:  1. Many citations are missing the year published. Please include this information and ensure that all citations are complete. Please also ensure appropriate capitalization in references cited (e.g., reference #14). 	 1. We have reviewed and updated all citations to include the year of publication and ensure their completeness. Additionally, we have addressed the capitalization issue in references cited, including reference #14. 2. We have carefully reviewed the citation formatting throughout the manuscript and made the necessary corrections to ensure consistency.

2. Citation formatting is inconsistent and should be corrected to be consistent.	
Tables and Supplemental Tables All acronyms used in tables (including table headings and within cells) should be defined in or under each table for convenient reference by readers.	We have ensured that all acronyms used in tables, including table headings and within cells, are defined in or under each respective table for the convenience of our readers.

VERSION 2 – REVIEW

REVIEWER	Standley, Krys University of Montana, Rural Institute
REVIEW RETURNED	08-Jan-2024

GENERAL COMMENTS	This feasibility and acceptability study used a mixed-methods approach which used random assignment to treatment and control groups and in which data from the quantitative and qualitative components were triangulated. The approach produced a rigorous study of the feasibility and acceptability of health coaching to support Type 2 Diabetes self-management in Saudi Arabia. The study also assessed preliminary health outcomes from the health coaching intervention. The authors concluded that high levels of feasibility and acceptability support advancement to a larger randomized controlled trial of health coaching for adults with Type 2 Diabetes in Saudi Arabia. The revisions made by the authors addressed most of my concerns from my initial review and have substantially improved the manuscript. However, some minor issues detract from the quality of the paper. I recommend that, with minor revisions, this paper will be suitable for publication. Detailed comments  1. Abstract: The authors may want to include an overview of the qualitative component of the study to round out the introduction and better orient readers to this mixed-methods study. As it is, the abstract is incomplete in describing the study 2. Introduction: Some acronyms still need to be spelled out upon first use. For example, SA, RCT (first used in the Introduction Section but first spelled out in the Materials and Methods Section), SPSS, STATA 3. Throughout: Some typos remain. Examples include “methods for recruitment, deliver and data collection,” “in several keyways,” 4. Methods: The two groups that participants were divided into are not well-introduced in this section. You state that “Participants were randomly assigned to one of two groups in a 1:1 ratio by an independent individual using a computer-generated random numbers system (SPSS), ensuring equal allocation chances,” but then proceed to the next section, “Intervention Delivery.” It would
---

	be helpful to state what the two groups were, (i.e., intervention and control) immediately after introducing the random assignment 5. Tables 5a. It is odd that, Supplemental Table 2 is listed in the narrative of the manuscript after Supplemental Tables 3, 4, and 5. Re-numbering these tables would allow for sequential placement in the text. 5b. Acronyms used in each table should be spelled out so that readers do not need to refer to the text to understand the content that is presented. 5c. Table 5 has been revised to include age and gender, which is an improvement. However, these additions create the need for an adjustment of the overall header of "Session #," which is no longer suitable across all of the columns. I suggest adding a row, titled "Session #," above only the numbered sessions. 6. Sample size: You state that "the statistician advised that for the upcoming main trial, a sample size calculation should be based on a clinically significant mean difference of 0.5%..." I am unfamiliar with the term "clinically significant mean difference." Should this be stated as a statistically significant mean difference instead? I also am curious about the 0.5% value listed for this mean difference. Is this meant to be 0.05, the typical p-value used to assess mean differences in the social sciences? If so, this value should have an additional zero and should not have a percentage sign associated with it. The phrase "clinically significant difference" is repeated later in this paragraph as well. 7. The two mentions of the joint display table are incorrectly labeled as Supplemental Table 5. In the Revised Supplemental Materials PDF, the joint display table is Supplemental Table 4. This mislabeling should be corrected.
--	---

VERSION 2 – AUTHOR RESPONSE

Reviewers' (1) comment	Our response
1. Abstract: The authors may want to include an overview of the qualitative component of the study to round out the introduction and better orient readers to this mixed-methods study. As it is, the abstract is incomplete in describing the study	We have now included an overview of the qualitative component in the abstract, providing a comprehensive description of the mixed-methods study as suggested (p.1).
2. Introduction: Some acronyms still need to be spelled out upon first use. For example, SA, RCT (first used in the Introduction Section but first spelled out in the Materials and Methods Section), SPSS, STATA	All acronyms have now been spelled out at first mention within the Introduction section for clarity (p.2-3).
3. Throughout: Some typos remain. Examples include "methods for recruitment, deliver and data collection," "in several keyways,"	Typographical errors have been corrected throughout the manuscript. This includes the correct phrasing of 'methods for recruitment, delivery, and data collection' and 'in several key ways' (p.3).

4. Methods: The two groups that participants were divided into are not well-introduced in this section. You state that “Participants were randomly assigned to one of two groups in a 1:1 ratio by an independent individual using a computer-generated random numbers system (SPSS), ensuring equal allocation chances,” but then proceed to the next section, “Intervention Delivery.” It would be helpful to state what the two groups were, (i.e., intervention and control) immediately after introducing the random assignment	We have clarified that participants were divided into an intervention group and a control group immediately after the random assignment description in the Methods section (p.5).
5. Tables	
5a. It is odd that, Supplemental Table 2 is listed in the narrative of the manuscript after Supplemental Tables 3, 4, and 5. Re-numbering these tables would allow for sequential placement in the text.	We appreciate the reviewer’s attention to detail. Upon review, we found that Supplemental Table 2 is appropriately mentioned after Table 1 in the manuscript (see p.3).
5b. Acronyms used in each table should be spelled out so that readers do not need to refer to the text to understand the content that is presented.	We have ensured that all acronyms within the tables are spelled out to make them self-explanatory.
5c. Table 5 has been revised to include age and gender, which is an improvement. However, these additions create the need for an adjustment of the overall header of “Session #,” which is no longer suitable across all of the columns. I suggest adding a row, titled “Session #,” above only the numbered sessions.	We have adjusted Table 5 by adding a row labeled 'Session #' above the session number columns for clarity.
6. Sample size: You state that “the statistician advised that for the upcoming main trial, a sample size calculation should be based on a clinically significant mean difference of 0.5%...” I am unfamiliar with the term “clinically significant mean difference.” Should this be stated as a statistically significant mean difference instead? I also am curious about the 0.5% value listed for this mean difference. Is this meant to be 0.05, the typical p-value used to assess mean differences in the social sciences? If so, this value should have an additional zero and should not have a percentage sign associated with it. The phrase “clinically significant difference” is repeated later in this paragraph as well.	Thank you for your comment and the opportunity to clarify these terms. The term "clinically significant mean difference" refers to the minimum change in a clinical outcome that is considered meaningful from a therapeutic or clinical perspective, which is distinct from "statistical significance." In our context, we are discussing the change in HbA1c levels, a clinical outcome measure for diabetes control. A 0.5% change in HbA1c is widely recognized as a clinically significant difference because it has implications for the risk of diabetes-related complications, as noted in various diabetes guidelines and literature. For example, the study by Lameijer et al., published in the Journal of Clinical and Translational

	Endocrinology in 2020, defines a clinically significant HbA1c decrease as a difference of ≥ 5 mmol/mol (0.5%) between baseline and the last available HbA1c concentration, according to the NICE guideline. This definition is consistent with the broader diabetes research community, which often recognizes a 0.5% change in HbA1c as a benchmark for clinical significance, as also evidenced by multiple studies cited in our literature review. The value of 0.5% should not be confused with the p-value used in statistical hypothesis testing.
The two mentions of the joint display table are incorrectly labeled as Supplemental Table 5. In the Revised Supplemental Materials PDF, the joint display table is Supplemental Table 4. This mislabeling should be corrected.	Thank you. It has been Revised (p.21)

VERSION 3 – REVIEW

REVIEWER	Standley, Krys University of Montana, Rural Institute
REVIEW RETURNED	14-Apr-2024
GENERAL COMMENTS	The authors have addressed all of my outstanding concerns.

VERSION 3 – AUTHOR RESPONSE